# SGXDump: A Repeatable Code-Reuse Attack for Extracting SGX Enclave Memory

HanJae Yoon 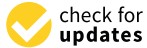 and ManHee Lee *

Laboratory of High Performance and Secure Computing, Department of Computer Engineering, Hannam University, Daejeon 34430, Korea; hanjae.karoha@gmail.com
* Correspondence: manheelee@hnu.ac.kr

**Abstract:** Intel SGX (Software Guard Extensions) is a hardware-based security solution that provides a trusted computing environment. SGX creates an isolated memory area called *enclave* and prevents any illegal access from outside of the enclave. SGX only allows executables already linked statically to the enclave when compiling executables to access its memory, so code injection attacks to SGX are not effective. However, as a previous study has demonstrated, Return-Oriented Programming (ROP) attacks can overcome this defense mechanism by injecting a series of addresses of executable codes inside the enclave. In this study, we propose a novel ROP attack, called SGXDump, which can repeat the attack payload. SGXDump consists only of gadgets in the enclave and unlike previous ROP attacks, the SGXDump attack can repeat the attack payload, communicate with other channels, and implement conditional statements. We successfully attacked two well-known SGX projects, mbedTLS-SGX and Graphene-SGX. Based on our attack experiences, it seems highly probable that an SGXDump attack can leak the entire enclave memory if there is an exploitable memory corruption vulnerability in the target SGX application.

**Keywords:** intel SGX; trusted computing; ROP; memory leak



## 1. Introduction

Intel's SGX is a widely-used technology to solve more fundamental security problems' [1–4] security by using hardware as the root of trust. It encrypts data and executables and stores them in a safe space: the `enclave`. It also prevents extraneous access to the enclave [5] by generating illegal memory access signals. In addition, SGX provides an advanced security model in that it protects the enclave even when the operating system is compromised.

However, SGX was not designed to defend against code-reuse attack, so its effectiveness depends on the security implemented by SGX application developers [6]. As a result, various studies have successfully demonstrated a code-reuse attack. ROP can be used at the stage of injecting the attack code in both micro-architectural timing attacks and controlled-channel attacks, which are two representative methods of attacking SGX. Micro-architectural timing attacks first record the access speed of hardware components such as the buffer, cache, input and output ports, as well as encryption engines. Then, they analyze any access time differences to infer protected data [7,8]. This attack often suffers from noise-related inaccuracies due to unpredictable system events and/or access speed variations. A controlled-channel attack is analyzing memory access patterns or memory contents through channels that an attacker can control such as page faults [9–14].

A typical code-reuse attack is a return oriented programming (ROP) attack. The ROP attack uses instruction sets called gadgets that are a series of instructions ending with *ret* instruction. Since all the gadgets are already contained in the victim binary, attackers do not need to inject new executable codes [15]. Furthermore, refs. [16,17] demonstrated that *BLX*, a branch instruction of ARM CPU, and *jmp* can be ending instructions to constitute

a gadget. In addition, refs. [18] proposed Sigreturn Oriented Programming (SROP) using the *restore_sigcontext()* function that is used to process signals in kernel mode for restoring stacks after a context switch. Refs. [19] proposed Blind ROP (BROP) that performs ROP attacks by analyzing binary's memory space features without direct access to the source code or binaries. Refs. [14,20–23] took the control flow of the process within enclaves by using the ROP. However, the methods proposed in previous studies require detailed analysis of the enclave software because it is essential to know how the software processes the protected data. This makes it very hard to carry out ROP attacks when the source code is not available. In addition, when the ROP payload is injected, a number of gadgets runs sequentially, making it difficult to perform complex tasks such as communication with the outside of the enclave.

In this paper, we propose an ROP payload that implements iterative and conditional statements without *cmp* instruction. By using this method, we constructed a communication channel to outside of the enclave. We developed the SGXDump ROP attack and the Probe Array Monitor (PAM) kernel module to prove that our implementation is working well. First, SGXDump in the form of an ROP payload is injected into the enclave software and reads the entire enclave memory. At each byte value, SGXDump accesses its corresponding page in the probe array to set the access bit of its page table entry to one. When the Probe Array Monitor detects this change, it translates the access bit's location to the value that SGXDump reads from the enclave's memory.

We summarize the main contributions of this study as follows:

- We present an ROP attack that implements loops inside the attack payload to leak enclave data. Our ROP attack utilizes page table entries' access bits so that the PAM can retrieve the enclave's memory.
- We give a real example of ROP code for attacking SGX. Unlike the usual ROP codes that simply call a specific function with designated parameters, SGXDump walks through the entire memory while accessing the probe array. We demonstrate how to build ROP code for implementing loops with limited gadgets. This will be useful for many researchers who want to devise ROP attacks on SGX.
- We demonstrate successful ROP attacks that could lead to a leak of the entire memory by using SGXDump and PAM to attack two well-known SGX open source projects: mbedTLS-SGX and Graphene-SGX. While mbedTLS-SGX is a very useful tool for SGX applications that need to use an SSL library for secure communication [24]. Graphene-SGX is a tool for running unmodified code on SGX [25]. Therefore, any vulnerable code found in either of these projects could be serious threats for SGX service owners and users.

We organize the rest of the paper as follows. Section 2 provides background related to our study. Section 3 explains the SGXDump threat model. Section 4 gives an overview of SGXDump and Section 5 gives a detailed account of how our attack is designed. In Section 6, we explain our attack implementation. After presenting related works in Section 7, we conclude in Section 8.

## 2. Background

**Intel SGX.** Intel Software Guard Extensions (SGX) is a set of instructions that allows a user process to create a secure area called an enclave inside its address space. An enclave has separate code and data sections including a stack and heap; SGX safeguards this enclave against external access [5,6]. However, as many previous studies presented, enclave's memory is still vulnerable to various attacks and the code-reuse attack such as ROP is demonstrated to be effective in leaking the enclave's memory [14,20]. In our study, we build the ROP payload that has a loop format by using gadgets inside the enclave. It should be noted that, SGX SDK libraries are statically linked to an enclave's code at build-time so that the enclave becomes a self-contained binary. This property may hamper an attacker's ROP composition because the target enclave binary should be available to the attacker beforehand. However, this is not necessarily true because SGX is designed to

secure enclaves under a compromised OS and the OS has the privilege of looking into the enclave binaries [5,6].

**Enclave ASLR.** Address space layout randomization (ASLR) is a technique that randomly selects the loading address of executables including libraries, the heap, and stack in the address space [26]. This is an effective countermeasure against memory corruption vulnerabilities (e.g., from buffer overflow) because injected code usually needs the target library function's address.

Although ASLR should be an effective defense against normal ROP attacks, our attack works well even if ASLR is used. In this study, an attacker with root privilege is able to find the base address of the enclave binary at each execution. Additionally, ASLR does not randomize the section address inside of the enclave. Therefore, once an attacker knows the base address of the loaded enclave, he can calculate all the addresses of the necessary ROP gadgets. SGX-Shield [27] addresses this limitation of enclave ASLR, but this approach is yet to be deployed in the official SGX library.

**Virtual Address Translation.** As a computer operates, virtual addresses need to be translated into physical addresses. For this purpose, the OS uses a per-process page table to manage its memory at the page level. The page table is an array of page table entries (PTE). A PTE contains the physical address of a virtual page and additional information including status bits such as accessed, present, rw, user, and dirty, among others. From these status bits, we make use of the access bit. When an instruction accesses a page, its access bit is set to one. Later, the access bit is used by the page replacement algorithm to select which page will be swapped out to the secondary disk [28,29].

When an attacker has root privilege, this access bit can be useful as a covert channel in the following way. First, a process allocates 256*4096-bytes for called a probe array (*pa[]*), which needs 256 consecutive pages for a page size of 4KB. Let us assume that the process acquires one byte of secret information, *s*. The unsigned integer value of *s* ranges from 0 to 255. Then, the process uses *s* for accessing *pa[s*4096]*. This memory reads or writes operation sets the *s*th page's access bit to one. When the attacker residing in the kernel monitors *pa[]*'s access bit, he can obtain *s* easily. In this paper, we will demonstrate how to develop the SGXDump ROP payload and the kernel module, both of which collaboratively leak the contents of the target enclave application's memory.

## 3. Threat Model

Figure 1 shows the overall threat model. The primary goal of SGXDump is to extract the memory's entire contents, including any code and data in the target enclave by using the ROP payload. To this end, we first suppose that the attacker is able to acquire root privilege, so has complete control of the OS. This is not an outlandish assumption because Intel SGX's programming model is designed to secure the enclave under the assumption that any of the software or hardware components, except the enclave, could be compromised [30]. Thus, we assume that the attacker can install a kernel module for checking the access bit of the victim process.

Since the prime attack vector for disclosing an enclave's memory in our approach is an ROP attack, we assume that the target application running the enclave has a memory corruption vulnerability (e.g., a stack buffer overflow). In some cases, the binary is encrypted at build time to further improve the security of the target enclave. This binary is decrypted just before the enclave is loaded, so it is practically impossible to make ROP payloads because pre-analysis of enclave binary is not feasible. Therefore, we assume that the target enclave is not encrypted at build time.

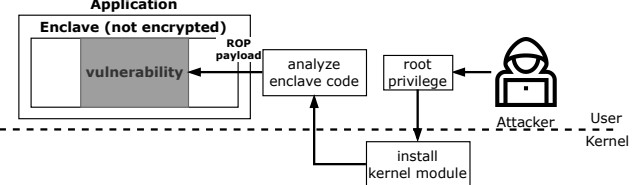

**Figure 1.** The threat model.

## 4. SGXDump Attack Overview

Figure 2 depicts the overall architecture of the SGXDump attack. Two main units are the SGXDump and Probe Array Monitor. SGXDump in the form of an ROP payload is injected into the vulnerable code and keeps reading one byte from the enclave memory, accessing one page of the probe array until it reads through the entire enclave memory. The Probe Array Monitor (PAM) module is inserted into the kernel by the attacker beforehand and waits for SGXDump's access to the probe array. The PAM translates the locations of access bits into enclave memory values. Finally, the attacker obtains the enclave data by reading the PAM's memory through *procfs*. In the following subsections, we will explain more about the SGXDump and PAM.

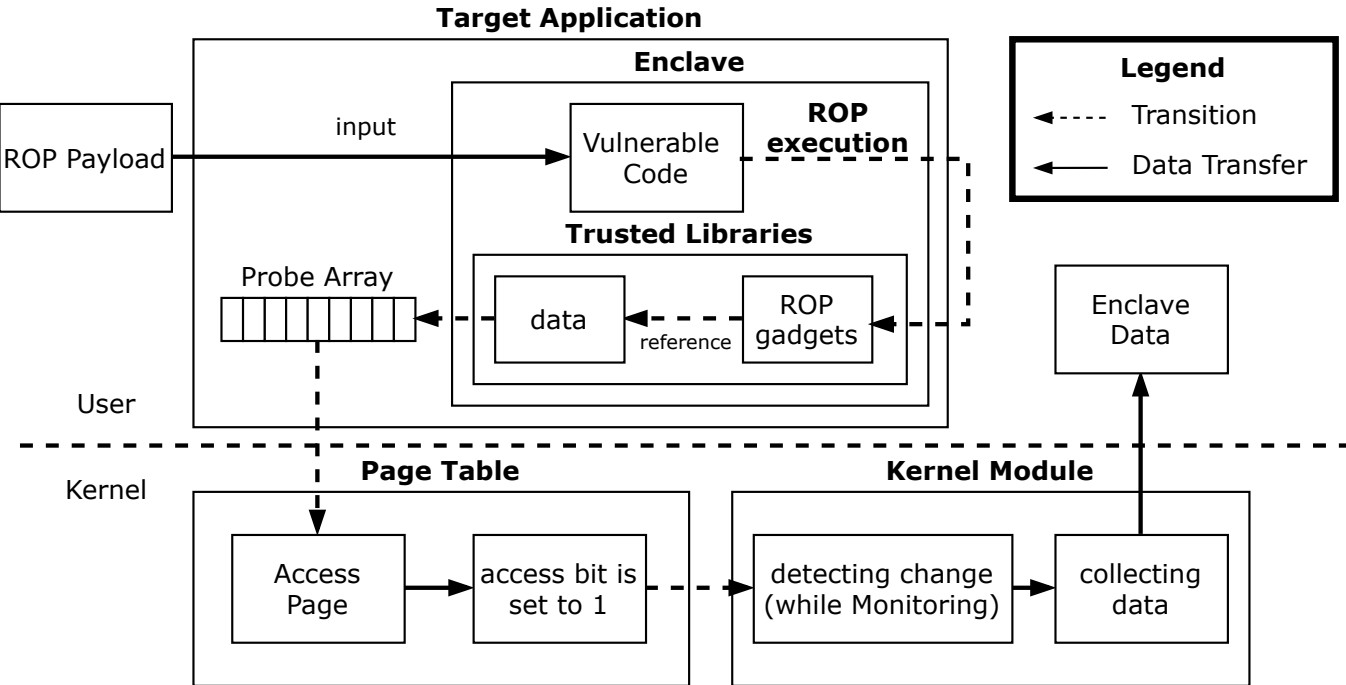

**Figure 2.** The proposed SGXDump framework.

### 4.1. SGXDump

As explained in Section 3, the typical role of the ROP payload is to escalate the privilege of the malicious code for running the shell or disabling the security enforcement (e.g., the SELinux or ASLR policy). Since the main task of these ROPs is to make system calls with proper parameters, its payload only needs several gadgets.

However, the SGXDump ROP payload has a rather complicated task: it extracts the entire enclave's memory including code and data using the probe array's page table entry (PTE). To this end, SGXDump turns enclave memory values into virtual page numbers (VPNs) in the accessed PTE as follows. It reads one byte from enclave memory and binary-shifts the value to the left twelve times, multiplying by 4096 (i.e., the page size). Then, it adds the base address of the probe array to the value and uses it as an index to access the probe array. This access leads to the access bit in the probe array's page table entry

(PTE) being set to one. Based on the index of the accessed probe array, the PAM can recover the value that SGXDump read from the enclave memory. To obtain the entire contents of the enclave's memory, SGXDump iterates this process until it reaches the end of the enclave's memory.

*4.2. Probe Array Monitoring*

The main task of the PAM is to detect a change in a PTE of the probe array and to recover the value by locating the accessed page number in the PTE. Then, PAM clears the access bit in the PTE and continues to wait for a change in the PTE. However, SGXDump is not able to detect the clearing event carried out by PAM, so it cannot decide when to read the next bytes. If the two components are not synchronized well, SGXDump could access the probe array multiple times before the PAM retrieves the information or vice versa, resulting in noisy or incorrect data leakage.

In order to make sure that the SGXDump and PAM leak data correctly, we devised a simple binary semaphore using four-byte memory in the target application's stack area. We call this memory a *sync*. Unlike a normal binary semaphore, which uses zero or one, our *sync* uses zero or an offset, which will vary depending on target enclave applications. Zero means that SGXDump can continue to read the next byte. After accessing the probe array, it writes an offset value at the *sync*. This offset is used to jump to the starting address of a waiting loop in the ROP payload of SGXDump, which will be explained in detail in the next section.

Simultaneously, the PAM keeps checking the *sync*. If the value stored in the *sync* is not zero, it means that SGXDump has finished accessing the probe array. Now, it is the PAM's turn to retrieve information from the PTE. After this, the PAM clears the access bit of the PTE and resets the *sync* to zero, indirectly signaling SGXDump to read the subsequent byte.

## 5. Attack Design

In this section, we describe the attack process in detail. We begin with memory selection for the probe array. Then, we explain gadget searching and the ROP payload structure. Finally, we finish this section by looking at data leakage and synchronization.

*5.1. Choosing the Probe Array*

The probe array is a trampoline memory array used to set the access bit in the page table entries (PTE). To minimize noise, we choose the memory area as a probe array that satisfies the following conditions.

- **Mapped memory area.** If an unmapped memory area is chosen as the probe array, unnecessary exceptions will occur whenever the ROP accesses the probe array. To avoid this situation, we select a mapped memory area for the probe array.
- **No access by other threads.** Our main idea is to detect the access bit of the probe array by executing ROP gadgets. If other threads access the probe array, this access will cause unwanted noise. Therefore, we choose a memory area that is not accessed by other threads.
- **Memory area outside of the Enclave.** Since SGXDump will read through all the enclave memory, access bits of the memory will change as a consequence. To avoid undesirable noise on the access bits, it is necessary to pick a memory area outside of the enclave.

Since we assume the attacker obtained the root privilege, the attacker can find a memory area that satisfies the above conditions as follows. The first step is to obtain the memory mapping information of the target process by analyzing the */proc/pid/maps* file as shown in the Figure 3. After checking the mapped memory area in lines 1~6 and the enclave memory area in lines 15~20, the attacker can find memory areas located outside of the enclave. The second step is to identify every thread that accesses the possible probe array memory area by using the debugger. Then, the attacker can select a memory area

that there are no threads accessing the area. In this research, we often selected one from the stack area, since it normally satisfied the above conditions.

### 5.2. Finding Gadgets for the ROP Payload

As explained in the Intel® Software Guard Extensions (Intel® SGX) SDK for Windows OS, SGX does not allow dynamic linking, meaning that all necessary functions' binaries are copied to the enclave code at build time [6]. Each enclave needs its own function sets at different locations. This is why it is almost impossible to construct a silver bullet ROP payload that will work for any enclave applications. Therefore, an adversary has to build a customized ROP payload for each target enclave.

```
1   5647291e0000-5647291e1000 r--p 00000000 08:02 21890951              /home/user/testApp
2   5647291e1000-5647291e2000 r-xp 00001000 08:02 21890951              /home/user/testApp
3   5647291e2000-5647291e3000 r--p 00002000 08:02 21890951              /home/user/testApp
4   5647291e3000-5647291e4000 r--p 00002000 08:02 21890951              /home/user/testApp
5   5647291e4000-5647291e5000 rw-p 00003000 08:02 21890951              /home/user/testApp
6   56472a852000-56472a894000 rw-p 00000000 00:00 0                     [heap]
7   7fe1ebd8a000-7fe1ebda6000 r-xp 00000000 08:02 23855405              /lib/x86_64-linux-gnu/libz.so.1.2.11
8   7fe1ebda6000-7fe1ebfa5000 ---p 0001c000 08:02 23855405              /lib/x86_64-linux-gnu/libz.so.1.2.11
9   7fe1ebfa5000-7fe1ebfa6000 r--p 0001b000 08:02 23855405              /lib/x86_64-linux-gnu/libz.so.1.2.11
10  7fe1ebfa6000-7fe1ebfa7000 rw-p 0001c000 08:02 23855405              /lib/x86_64-linux-gnu/libz.so.1.2.11
11  7fe1ebfa7000-7fe1ec1f7000 r-xp 00000000 08:02 8134970               /usr/lib/x86_64-linux-gnu/libprotobuf.so.10.0.0
12  7fe1ec1f7000-7fe1ec3f7000 ---p 00250000 08:02 8134970               /usr/lib/x86_64-linux-gnu/libprotobuf.so.10.0.0
13  7fe1ec3f7000-7fe1ec3ff000 r--p 00250000 08:02 8134970               /usr/lib/x86_64-linux-gnu/libprotobuf.so.10.0.0
14  7fe1ec3ff000-7fe1ec400000 rw-p 00258000 08:02 8134970               /usr/lib/x86_64-linux-gnu/libprotobuf.so.10.0.0
15  7fe1ec400000-7fe1ec401000 r--s 00000000 00:06 548                   /dev/isgx
16  7fe1ec401000-7fe1ec47e000 r-xs 00001000 00:06 548                   /dev/isgx
17  7fe1ec47e000-7fe1ec483000 r--s 0007e000 00:06 548                   /dev/isgx
18  7fe1ec483000-7fe1ec484000 r--s 00083000 00:06 548                   /dev/isgx
19  7fe1ec484000-7fe1ec587000 rw-s 00084000 00:06 548                   /dev/isgx
20  7fe1ec587000-7fe1ec597000 ---s 00187000 00:06 548                   /dev/isgx
21  7fe1ec597000-7fe1ec5d7000 rw-s 00197000 00:06 548                   /dev/isgx
22  7fe1ec5d7000-7fe1ec5e7000 ---s 001d7000 00:06 548                   /dev/isgx
23  7fe1ec5e7000-7fe1ec5ea000 rw-s 001e7000 00:06 548                   /dev/isgx
24  7fe1ec5ea000-7fe1ec5fa000 ---s 001ea000 00:06 548                   /dev/isgx
25  7fe1ec5fa000-7fe1ec5fb000 rw-s 001fa000 00:06 548                   /dev/isgx
26  7fe1ec5fb000-7fe1ec600000 ---s 001fb000 00:06 548                   /dev/isgx
27  7ffeff234000-7ffeff255000 rw-p 00000000 00:00 0                     [stack]
28  7ffeff342000-7ffeff345000 r--p 00000000 00:00 0                     [vvar]
29  7ffeff345000-7ffeff346000 r-xp 00000000 00:00 0                     [vdso]
30  ffffffffff600000-ffffffffff601000 --xp 00000000 00:00 0             [vsyscall]
```

**Figure 3.** Contents of */proc/pid/maps* file.

During the attack preparation step, the adversary will search for a set of gadgets. We use ROPgadget [31] for finding all gadgets that can be used to build ROP payloads. Since certain library functions are likely to be included in many enclaves, it is usually better to use gadgets from SGX's trusted library. However, as we mentioned above, each enclave uses its own set of library functions, so there may be cases in which gadgets that the adversary wants to use are not available in any of the enclave's binary code. We discovered that there are various combinations of gadgets that perform the same operation. For example, one gadget, *inc addr*, in mbedTLS-SGX, increments the value stored at *addr*. However, Graphene-SGX does not have that gadget. Instead, we used six other gadgets and composed an ROP sequence shown in the first nine lines of Listing 4, which performs an equivalent operation to *inc addr*. In this fashion, we successfully built the SGXDump ROP payloads for mbedTLS-SGX and Graphene-SGX, despite them having completely different sets of gadgets. To demonstrate that the SGXDump ROP payload can be created in other SGX applications, we constructed payloads in two additional SGX applications, Asylo [32] and Mystikos [33], and presented their gadgets in Appendix A. Based on our experience of finding gadgets from four SGX applications, it is very likely that SGXDump ROP payloads can be constructed for most enclave applications.

Before going further, we make the case that our attack can work well even when address space layout randomization (ASLR) is enabled. In an ASLR environment, the same enclave is loaded at different base addresses, causing a gadget address found in one execution to change in the next execution. However, it is worth noting that while base addresses are random at every execution, the relative addresses of the text and data sections inside the enclave memory do not change. This means that a gadget's distance from the base address is always the same. Therefore, a new version of the ROP payload can be constructed correctly when the base address of the target enclave becomes available.

**Identifying the Enclave's Base Address.** To find out the base address of the Enclave, we make use of the */proc/pid/maps* file. This file contains the address information of the */dev/isgx* file, which has the device name of the SGX kernel module [34]. The address is identical to the base address of the loaded enclave in the target process.

*5.3. SGXDump ROP Payload Structure*

This subsection describes the SGXDump ROP payload in detail. Since the SGXDump consists of a relatively large amount of code compared to normal ROP payloads, we grouped its code into four steps and depicted the overall flow in Algorithm 1. After giving a short summary of each step, we provide in-depth explanations.

---

**Algorithm 1** Flow of ROP payload.

---

　1: *Registers* ← ∅
　2: **while** true **do**
　3:　　Calculate sync address (Step 1)
　4:　　**if** *RSP* does not have the next gadget address **then**
　5:　　　*continue*
　6:　　**end**
　7:　　Get secret data (Step 2)
　8:　　Shift and access probe array (Step 3)
　9:　　Loop (Step 4)
　10: **end**

---

- **Step 1. Calculating sync address.** The ROP payload initializes several registers to avoid unpredictable behaviors. After reading the value at the sync address, SGXDump uses the value as a sign to access the next data or to jump back to check the sync address for busy waiting.
- **Step 2. Getting protected data.** At this stage, the ROP retrieves two values from the index address and the target address and adds the two values to calculate the next address in enclave memory. Then, the ROP reads one byte from the address and stores it in a register temporarily.
- **Step 3. Shifting protected data and accessing probe array.** The ROP payload left-shifts the data 12 times, multiplying by 4096, and uses the resulting value as an index to access the probe array.
- **Step 4. Looping.** Before looping back to Step 1, the ROP payload increments the content at the index address and writes an offset value at the sync address to make Step 1 perform a busy waiting.

A detailed description of each step is given in the following.

**Step 1. Initializing and Calculating address.** The attacker initializes the registers to be used for the ROP payload to 0. Please note that Line 1 in Listing 1 is not a single instruction, but a series of gadget addresses, each of which is a register reset instruction followed by a `ret` instruction.

Next, the attacker loads the address of the sync to `reg1` since the `pop reg1` instruction takes the top of the stack (currently, *sync_address*) and stores it to `reg1`; an example of the ROP payload is shown in Figure 4. It is worth mentioning that the actual content in Line 2 of the ROP payload is not an instruction but a gadget address whose instruction is `pop reg1` followed by a `ret` instruction, as in Figure 4a. We chose this way for better understanding and convenience. Thus, to construct a real SGXDump ROP payload, all instructions in the following Figures and Lists will be substituted with their corresponding gadgets' addresses.

In Figure 4a, executing the `ret` instruction makes the control flow return to the address that is loaded in the `rsp` register. The `rsp` register has the next gadget address. After returning, the next gadget (`pop reg1`) is executed as shown in Figure 4b. After executing `pop reg1`, the address of the sync is loaded in `reg1`, as shown in Figure 4c.

Then, Line 4 in Listing 1 loads the sync data to `reg1`. At first, the sync data is set to zero, so `rsp` register does not change after subtracting `reg1`. Since `rsp` register points to the current stack top, the succeeding `ret` instruction in the gadget makes the control flow go to Line 1 in Listing 2. If `reg1` is not zero, `sub rsp, reg1` would make the control flow jump back to somewhere in the stack. In our attack, we deliberately select this offset value of `reg1` to make the address jump to Line 2 in Listing 1, completing a busy waiting loop that is checking the content of the sync address. When the PAM module resets the sync data to zero, this busy waiting loop is broken and the control moves forward to Step 2.

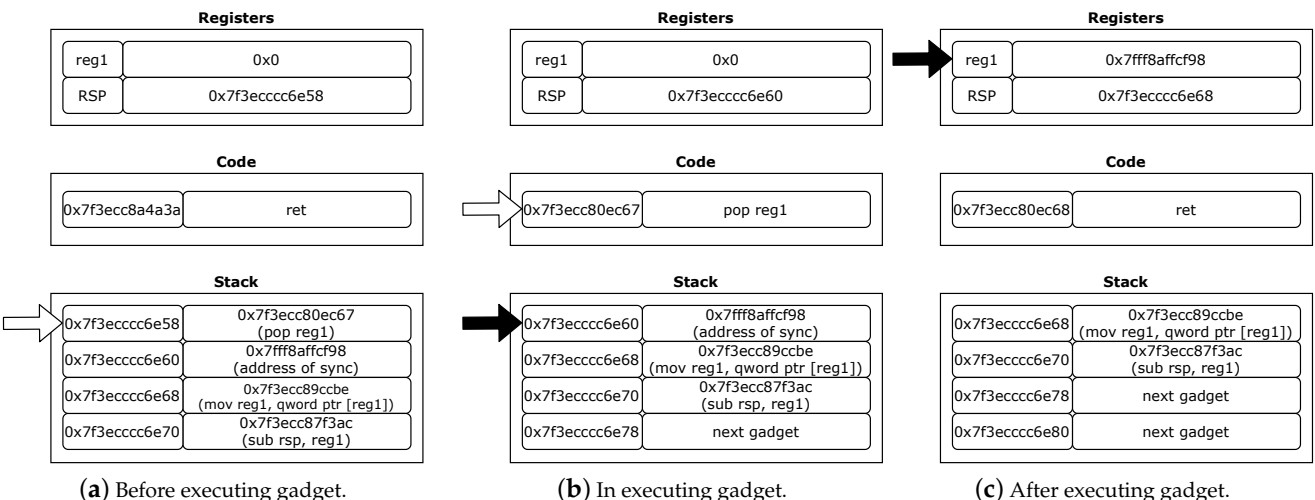

**Figure 4.** State of stack.

**Listing 1.** ROP payload—Step 1.

```
pop reg1
sync_address
mov reg1, qword ptr [reg1]
sub rsp, reg1
```

**Step 2. Obtaining protected data.** SGXDump first moves *index_address* to `reg2` as shown by Line 1 and Line 2 in Listing 2. Then, the attacker loads the data from *index_address* and writes it to `reg2` (Line 3 in Listing 2). We use this data as an index from the base address to point to a location in the enclave memory to leak next. Since *target_address* contains the base address of the target enclave, the addition of `reg3` and `reg2` (Line 6 in Listing 2) completes the memory address from which to leak data. Finally, SGXDump loads the data from the enclave and stores it in `reg2` (Line 7 in Listing 2).

**Listing 2.** ROP payload—Step 2.

```
pop reg2
index_address
mov reg2, byte ptr [reg2]
pop reg3
target_address
add reg3, reg2
mov reg2, byte ptr [reg3]
```

**Step 3. Shifting protected data and accessing probe array.** After collecting the protected data, the SGXDump uses the `shl` instruction to left-shift the data stored in `reg2` by *0xC* (Line 1 in Listing 3). After loading *probe_array_address* into `reg3`, the addition of `reg2`

and `reg3` creates an address that falls into the `reg2`th page of the probe array, assuming the page size is 4KB. Next, SGXDump accesses a page of the probe array through that address. Please remember that the value stored in `reg1` (Line 5 in Listing 3) is of no interest because the only reason for accessing the probe array is to set the access bit of its corresponding PTE to one.

Figure 5 shows an example of how an access bit is changed. When the gadget at Line 7 in Listing 2 accesses protected data at a certain enclave address, *0x1ba6d000*, it acquires *0x7f*, and left-shifts it by *0xC*. As a result, the protected data becomes *protected data * 4096, 0x7f000*. Since the start address of the probe array is *0x7ffd6c316000*, the addition of `reg2` and `reg3` gives *0x7ffd6c395000*. Line 5 in Listing 3 makes a read access at this address, setting the access bit of the page table to 1.

**Listing 3.** ROP payload—Step 3.

```
shl  reg2 ,  0xC
pop  reg3
probe_array_address
add  reg2 ,  reg3
mov  reg1 ,  byte  ptr  [reg2]
```

**Step 4. Looping.** Next, the SGXDump increments the value stored at *index_address* (Line 1–9 in Listing 4). Although this results in the same effect as using a certain gadget, `inc addr`, we would like to present an example showing that there are many ways of using the gadgets available to perform the desired operation when a specific gadget for that operation is not available.

After this, the SGXDump stores an *offset* value at *sync_address* (Line 10–14 in Listing 4). This offset is the difference of two addresses, Line 2 and 5 in Listing 1, which is used to make Step 1 continue to check the sync address. The final step is to move the control flow into Line 2 of Listing 1 so that SGXDump waits for the PAM to retrieve the protected data from the page table and to reset the value at the sync address to zero. This is why *rsp_address* has the address of Line 2 in Listing 1 and why the attacker loads it into the `rsp` register (Line 15 and 16 in Listing 4).

**Listing 4.** ROP payload—Step 4.

```
pop  reg1
index_address
mov  reg1 ,  byte  ptr  [reg1]
pop  reg2
0x1
add  reg1 ,  reg2
pop  reg2
index_address
mov  byte  ptr  [reg2] ,  reg1
pop  reg2
sync_address
pop  reg3
offset
mov  byte  ptr  [reg2] ,  reg3
pop  rsp
loop_start_address
```

*5.4. Synchronization and Data Leak via PAM*

We implemented the PAM using PTEditor [35]. PTEditor provides an API to set or clean the access bit of a page. We first explain how PAM synchronizes with SGXDump; later, we will demonstrate how PAM leaks data via the page table.

If SGXDump runs faster than PAM, it might access the probe array multiple times before the PAM clears the page table's access bits. This is why we proposed the use of four-byte memory, which we call *sync*, for synchronization between SGXDump and the PAM. PAM's implementation is much simpler than that of the SGXDump because the PAM was developed in C, whereas SGXDump is programmed using the return-oriented technique. Algorithm 2 depicts the PAM's flow. It first monitors whether the sync address contains a non-zero value. A non-zero value indicates that SGXDump has finished one iteration and is now in the state "*busy waiting*" until content of the sync address becomes *0x0*. After reading the protected data, the PAM clears the access bit of the PTE and changes the content of the sync address to *0x0*, thus signaling the SGXDump to continue reading the next bytes.

Finally, we exhibit an example to explain how PAM obtains data from the PTE. In Figure 5, SGXDump accessed the probe array at *0x7ffd6c395000*, setting the access bit of *0x7ffd6c395000*'s page table entry to one. Since the attacker knows the starting address of the probe array, *0x7ffd6c316000*, he can calculate the difference between the two pages, *0x7ffd6c395000 − 0x7ffd6c316000 = 0x7f000*. Right-shifting *0x7f000* twelve times generates the final piece of protected data, *0x7f*, originally stored in enclave memory address at *0x1ba6d000*. The PAM keeps collecting data via the probe array until the attacker can retrieve entire memory data from the target enclave.

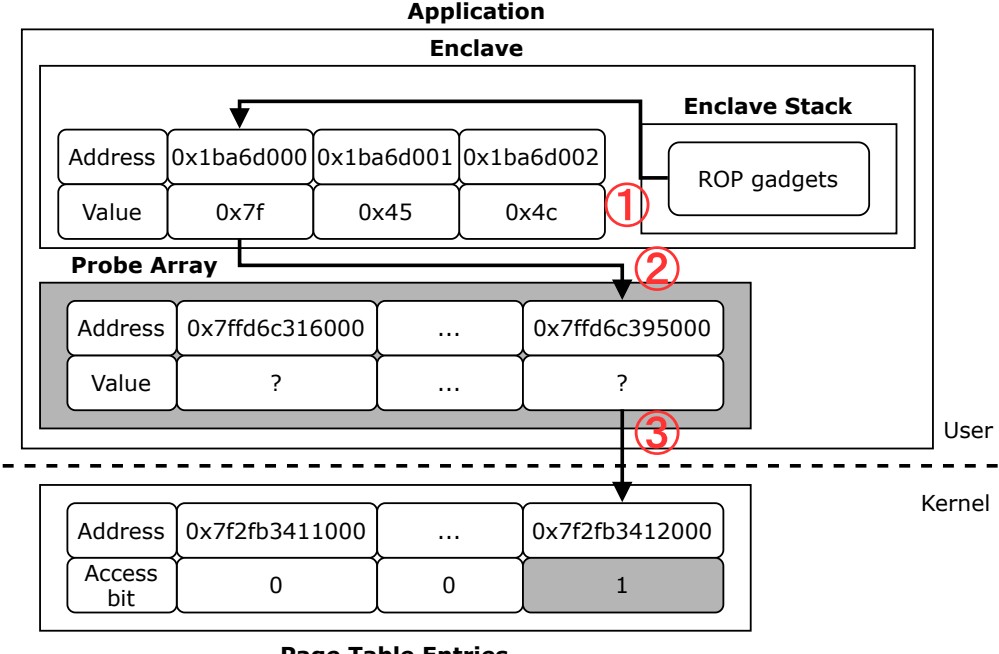

**Figure 5.** Flow of access bit change.

---

**Algorithm 2** Flow of Kernel module.

---

1: **while** true **do**
2:     Monitor the *address* of offset
3:     **if** $0x0$ is stored in *address* **then**
4:         *continue*
5:     **end**
6:     Get the address of the page with access bit is set to 1
7:     Clear all of the access bits
8:     Write $0x0$ to *address* of offset
9: **end**

---

## 6. Implementation

To verify the effectiveness of our attack, we carried out two attacks targeting SGX open source projects: mbedTLS-SGX, an enclave that provides various encryption and certificate-related algorithms [24], and Graphene-SGX, an enclave that helps non-SGX programs to be run in SGX [25].

### 6.1. mbedTLS-SGX

TLS communication can be insecure when an attacker with root privilege accesses the communication process' memory. mbedTLS-SGX was introduced to overcome this limitation by preventing the compromised OS from accessing the contents of a TLS communication running inside an enclave.

Figure 6a depicts how an mbedTLS-SGX-supported client works when communicating with a remote server and how an SGXDump attack is carried out. We did not draw detailed behaviors from TLS communications to gain a better understanding of how to implement the SGXDump attack.

After exchanging session keys with a remote server, mbedTLS-SGX continues to access plaintext and ciphertext for encryption and decryption before/after sending and receiving ciphertexts. When terminating the TLS communication, it calls *close_connection*. At this point, we injected our SGXDump ROP code. Table 1 shows the gadgets we used in our attack. Please note that this does not mean mbedTLS-SGX's function has a vulnerability. We modified its source code to make it vulnerable to a buffer overflow attack.

**Table 1.** Gadget list used in mbedTLS-SGX.

| Gadgets | Function | Offset |
|---------|----------|--------|
| adc ecx, dword ptr [rax − 0x7d]; ret; | init_global_object | 0x19be1 |
| add cl, byte ptr [rax − 0x7d]; ret; | __vfprintf | 0x80f92 |
| add eax, ecx; ret; | get_dynamic_layout_by_idt | 0x17388 |
| add rax, rcx; ret; | get_dynamic_layout_by_idt | 0x17387 |
| inc dword ptr [rcx + 1]; ret; | __strtodg | 0xa4a37 |
| mov eax, dword ptr [rax + 8]; ret; | get_enclave_size | 0x18778 |
| mov qword ptr [rdi + 0x10], 0; ret; | typedef basic_string | 0x9228f |
| mov qword ptr [rdi], rax; ret; | _exception | 0x91467 |
| mov qword ptr [rdx], rax; xor eax, eax; ret; | elf_tls_info | 0x195d3 |
| mov rdi, qword ptr [rdi + 0x68]; ret; | _Ux86_64_setcontext | 0x9ccbe |
| nop; ret; | free | 0x861ef |
| pop rax; ret; | _ULx86_64_r_uc_addr | 0x9c4a3 |
| pop rcx; ret; | __dtoa | 0x8842d |
| pop rdi; ret; | __find_arguments | 0x0ec67 |
| pop rdx; pop rcx; pop rbx; ret; | do_egetkey | 0xa8a33 |
| pop rsp; ret; | _trts_ecall | 0x17d84 |
| shl eax, 0xc; ret; | get_heap_min_size | 0x18888 |
| sub rax, 1; ret; | _ZNKSt3__112basic_... | 0x92831 |
| sub rsp, rdi; mov rax, rsp; jmp rdx; | alloca | 0x7f3ac |

### 6.2. Graphene-SGX

To enhance security, only statically linked binaries can run in the enclave. However, since most applications prefer dynamic linking for improved memory efficiency, it is hard to use existing binaries with SGX. To overcome this limitation, Graphene-SGX was proposed to run non-SGX applications on SGX without the need for modification [25].

After creating an enclave, Graphene-SGX first loads the standard C libraries and necessary user libraries; it then loads one C application into the enclave, as shown in Figure 6b. The C application is able to run smoothly inside SGX by calling library functions and accessing memory within the enclave.

In theory, even when a binary running with Graphene-SGX has a vulnerability, SGX should protect its memory from illegal access. To show that an SGXDump attack can break

this protection, we implemented a simple ftp client with a buffer overflow vulnerability. We inserted the vulnerable code into the close function. In our experiment, the SGXDump attack successfully retrieved all data stored in the enclave's memory, as shown in Figure 6b.

It is of interest to note that Graphene-SGX is a relatively easy target for a SGXDump attack. Graphene-SGX always loads the C library, so there are plenty of gadgets with which to compose an attack. Table 2 shows the gadgets we used in our attack. Furthermore, the same C library is loaded for all the different applications run, so additional efforts to find new gadgets are not necessary.

**Table 2.** Gadget list used in Graphene-SGX.

| Gadgets | Function | Offset |
|---|---|---|
| add cl, byte ptr [rax − 0x77]; ret; | _IO_wstr_seekoff | 0x77fb1 |
| add eax, ecx; ret; | _fitoa_word | 0x4f2cd |
| add ecx, dword ptr [rax + 0x29]; ret; | _int_free | 0x85521 |
| add rax, rcx; ret; | __memchr_sse2 | 0x963a8 |
| add rax, rsi; ret; | __memrchr_sse2 | 0x96bcc |
| mov qword ptr [rdi + 8], rax; ret; | _IO_switch_to_main_get_area | 0x806b6 |
| mov qword ptr [rdx], rax; ret; | __GI___ctype_init | 0x32c4c |
| mov rax, qword ptr [rax]; ret; | __GI___res_state | 0x11a258 |
| movzx eax, byte ptr [rax]; ret; | __getc_unlocked | 0x7d352 |
| pop rax; ret; | mblen | 0x3cdb8 |
| pop rcx; pop rbx; ret; | _getopt_internal | 0xe2bea |
| pop rdi; ret; | init_cacheinfo | 0x25522 |
| pop rsi; ret; | strip | 0x25e42 |
| pop rsp; ret; | check_one_fd | 0x25c1a |
| shl rsi, 4; lea rax, [rdi + rsi + 8]; ret; | inet6_rth_getaddr | 0x11654b |
| sub esp, edi; dec dword ptr [rax + 0x39]; ret; | __tzfile_read | 0xbac9d |
| xchg eax, edi; ret; | strfromd | 0x3ff8e |
| xchg eax, esi; ret; | check_node_accept | 0xd65d9 |

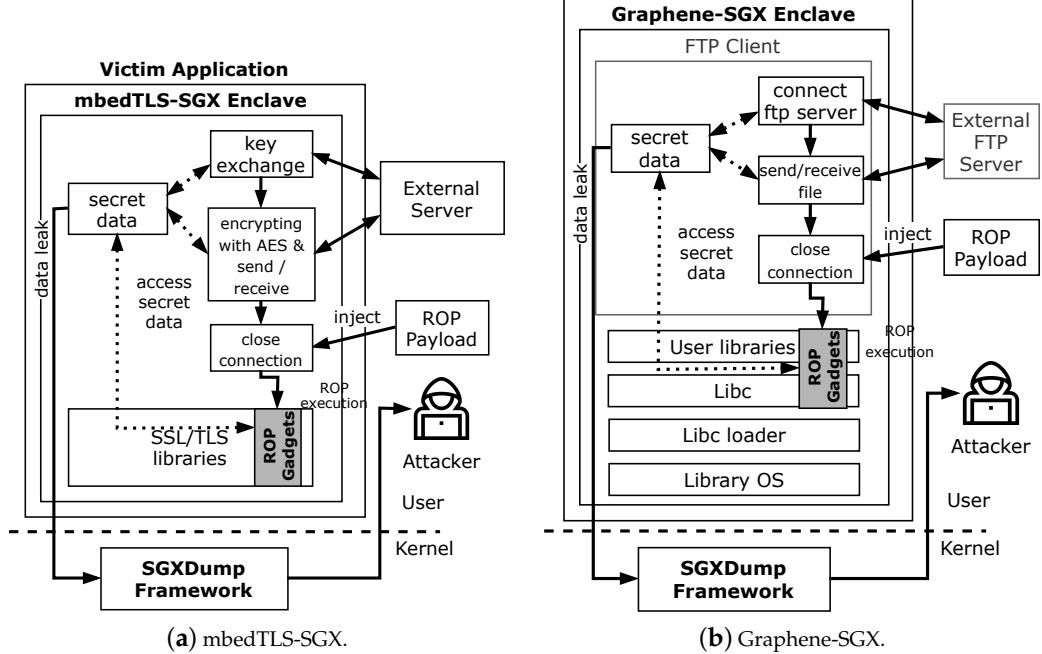

(**a**) mbedTLS-SGX.

(**b**) Graphene-SGX.

**Figure 6.** Attack model.

*6.3. Evaluation*

To evaluate our attacks, we compared our attack with other SGX attacks that mentioned the time that is taken to leak data and we calculated leaked bit per second (bps). Other attacks are summarized in the next section. The Table 3 shows the comparing results. As you can see from the table, in other attacks, they required too many attempts to leak data from the victim enclave and have low bps. They must try many times to leak data since their attacks have noise. Moreover, many times of attempts to leak data can affect bps.

In MemJam [36], they tried to recover the 128-bit AES key from SGX. Their non-optimized attack needed 2,000,000 observations to recover the key and took 5 min with noise. Since they needed m-arch contention and many times of observations, leaking bps could be low. In CopyCat [9], they tried to recover the 2048-bit RSA key from SGX. On average, they needed 39,400 steps and 20 s to recover the key with noise. CopyCat can leak data faster than MemJam since they do not need any m-arch contention. However, CopyCat's bps is lower than our attack.

In our attack, when we already found gadgets that are needed to construct the ROP payload, we can extract data with extremely high bps without noise with only one attempt. We tried to leak 8192-bit data by using SGXDump to our implementations. First, in mbedTLS-SGX, we can leak data in only one try without noise and it took 3 s on average. Next, in Graphene-SGX, as in the previous case as, we can leak data in only one try without noise and it took 12 s on average. In the case of Graphene-SGX, it needed a kernel patch to run the Graphene-SGX application, and this may affect bps.

**Table 3.** Evaluation result.

| Attacks | Type | Victim | Bit per Sec | Number of Attempts | Noise |
|---|---|---|---|---|---|
| MemJam [36] | m-arch contention | Data | 0.42 | 2,000,000 | Yes |
| CopyCat [9] | Ctrl channel | Control flow | 102 | 39,400 | Yes |
| SGXDump (mbedTLS-SGX) | Ctrl channel | Code and data | 2730 | 1 | No |
| SGXDump (Graphene-SGX) | Ctrl channel | Code and data | 682 | 1 | No |

## 7. Discussion

Side-channel attacks can be used to leak code and data from higher-privileged execution environments (e.g., an OS kernel, a hypervisor, an SGX enclave, or a trusted OS in the Secure World). Among these secure execution environments, Intel SGX has become widely used because it helps developers easily implement their own code on top of TEE. Owing to this growing popularity, attackers are starting to pay more attention to compromising SGX enclaves via side-channel attacks.

Table 4 summarizes the side-channel attacks that can be made against Intel SGX. In the same way as conventional side-channel attacks, we classify side-channel attacks against Intel SGX into three different categories according to the data acquisition channels: page table, branch prediction unit, and cache attacks.

Page table side-channel attacks are relatively convenient because the attacker can detect when the victim memory is accessed (e.g., enclave memory or the probe array in SGXDump) via architectural faults or via bit changes in PTE. Thus, many previous works have leveraged the access bit and present bit in PTE [10,12]. The access bit is used to determine a given page's eviction for cache management by the OS. Since a change in the PTE's access bit does not create an architectural fault, an adversary is required to keep polling this bit to detect memory access. On the other hand, a change in the PTE's present bit raises a page fault exception so that an attacker can detect a memory access event in a noise-free manner. As a variant of page table based-attack, Off-Limits [37] uses the segmentation carried out by Intel CPUs, which is applicable only in the x86 mode. A segmentation unit issues a general protection fault when a victim process accesses the memory area beyond the segmentation limit stored in the descriptor table.

**Table 4.** Comparison of the side-channel attacks against Intel SGX.

| Attacks | Victim | Data Acquisition Channel | Noise | Root Privilege | Granularity | Attack Precondition |
|---|---|---|---|---|---|---|
| SGXDump | Code and data | PTE's access bit | No | Yes | 1 Byte | Binary code, overflow vulnerabilities |
| Foreshadow | Code and data | PTE's present bit, Cache timing | Yes | No (mprotect) | 1 Byte | Secrets in the L1 cache |
| CopyCat | Control flow | APIC timer manipulation, the number of executed instruction | Yes | Yes (Loadable kernel module for APIC event call back) | Instruction | (Compiled) source code |
| Nemesis | Control flow | APIC timer manipulation, Interrupt latency | Yes | Yes | Instruction within one cache line | (Compiled) source code |
| MemJam | Data | 4K Aliasing, False dependency of memory read-after-write event, Cache line timing | Yes | Yes | Word or cache line | Virtual memory offset of critical data |
| Single Trace | RSA key generation code | PTE's present bit | No | Yes (SGX-Step) | Page | (Compiled) source code |
| Off-Limits | Control flow, one instruction | Segmentation and paging | No | Yes | Page | (Compiled) source code, only 32-bit mode |
| BranchScope | Control flow | Pattern history table | Yes | Yes | Instruction | Virtual address of victim's code |
| Bluethunder | Control flow | Global history register | Yes | Yes | Instruction | (Compiled) source code |
| Branch shadowing | Control flow | Branch prediction | Yes (Unconditional) No (Conditional and indirect) | Yes | Instruction | (Compiled) source code |

Modern CPUs have branch prediction units that consist of a branch target buffer and the directional predictor. BranchScope [38] and BlueThunder [39] leverage 1-level and 2-level predictors in the directional predictor for recording whether previous branch instructions are carried out. They generate intentional collisions with the directional predictor in the same physical core to disclose the result of executed branch instructions. Lee et al. presented the branch shadowing [40] technique that abuses the branch prediction unit by using the shadow code, which is aligned with the victim code in the enclave. After running the victim code, an attacker runs the shadow code and measures the elapsed clock cycles. Based on this measurement, the attacker infers whether the executed branch instruction in the victim enclave has been taken or not.

Cache timing measurement attacks are traditional, but still useful side-channel attacks. With this approach, the main task of the attacker is to measure the elapsed latency when accessing the target memory. MemJam [36] exploits the false dependencies of memory read-after-write events in the same physical core. A CPU reads memory with a virtual address, but the L1 cache tags its contents with the physical address. Therefore, when several processes access the memory with the same virtual address in the same physical core, the L1 cache cannot determine if this access is a hit or a miss ahead of the address translation. MemJam measures this address translation latency in order to disclose the target memory contents.

MicroScope [41], Stacco [42], and SGX-Step [13] were proposed as useful frameworks assisting side-channel attacks against enclaves. For example, CopyCat [9], Nemesis [43], and Single Trace [12] observe the single-stepping trace from a victim enclave on top of SGX-Step [13].

The overall structure of SGXDump is analogous to that of Foreshadow [10]. These have three similar components: a secret reference code, reference buffer, and PTE properties. In both approaches, the secret reference code dereferences the enclave memory and accesses an element in an array of 256 (4 KB) pages. The secret value is used to decide which element of the reference buffer is referenced. After access to the element, both detect changes in a PTE property (i.e., the access bit and the present bit). The key difference is how the secret reference code is created. In SGXDump, we create the secret reference code, i.e., the ROP payload using ROP gadgets that reside in the enclave, whereas Foreshadow uses small code snippets outside the enclave to execute transient instructions.

Unlike the aforementioned side channel attacks used against Intel SGX, SGXDump leverages the PTE's access bit as a covert channel. This is because SGXDump requires a specific way (e.g., software vulnerabilities) to leak the enclave's memory via a covert channel. For example, SGXDump exploits a stack overflow vulnerability to inject its ROP payload and executes the ROP gadgets to disclose the enclave's contents through a PTE-based covert channel to prove our ROP payload is well working. Even though there is a limitation in that a target enclave must contain vulnerabilities to run the ROP gadgets, SGXDump has the advantage of being able to nullify side channel attack countermeasures [44]. Previous research into other attacks has [14,45] also relied on software vulnerabilities.

Intel SGX can effectively prevent illegal memory access and code injection attacks. Since it provides such secure execution environment even when an attacker has full control of the OS, it has become widely adopted [5]. Owing to this growing popularity, attackers are starting to pay more attention to compromising SGX enclaves via the code-reuse attack.

Before constructing the payload for the ROP attack, it is generally necessary to statically analyze the binary to obtain the address of the gadget required for the payload. However, this method has the disadvantage that it is not possible to obtain the address of the gadget if the binary is encrypted or access to the binary is prohibited. In Dark-ROP [14], they proposed a method to find the gadget necessary for ROP attack by using the leaf function of SGX when the enclave is encrypted.

In order to perform an ROP attack, a memory vulnerability is required to inject the attack payload because it could not overwrite the return address with an ROP payload. For this, the memory layout of the victim process is analyzed through a large number of code probing, and in the case of SGX, the root privilege is necessary. Such code probing

provides an opportunity for the victim process to detect the attack and sometimes causes the victim process to crash and terminate unexpectedly. In [21], the authors proposed a novel method for an ROP attack without a root privilege and a crash by using *ORET* and *CONT* instructions of enclave. The instructions are used when a context switch happens from a kernel context to an enclave context after the kernel handled an exception occurring inside the enclave. They also minimized the crash of victim processes by inserting fake stacks into the switched context. A similar approach was used in [22].

In [23], the authors injected the ROP payload using the re-entry vulnerability of AEX. When an exception occurs in the enclave, the SGX enabled processor exits the enclave by using Asynchronous Enclave Exit (AEX). At this time, if the value of the register is left as it is, an attacker can leak the value of the register of the process being executed in the enclave, so AEX saves the value of the current register in the State Save Area (SSA) before exiting the enclave and replaces the registers with synthetic values. After handling the exception, SSA is used to restore the context before re-entering the enclave. The authors demonstrated that by manipulating the SSA at this time, it is possible to inject an ROP payload without a memory vulnerability. Through this attack, they were able to exfiltrate the enclave's data through memcpy.

## 8. Conclusions

In this paper, we proposed an ROP payload that can loop inside the payload by using gadgets in the enclave. To demonstrate the effectiveness of our method, we implemented a SGXDump that can extract an enclave's entire memory. Once a memory corruption vulnerability is found in a target SGX application, an SGXDump ROP payload can be injected. Its main role is to read bytes from enclave memory and access pages of a probe array in which we use those bytes' values as indexes. Meanwhile, a previously installed kernel module, the PAM (Probe Array Monitor), keeps monitoring the change of access bits of the probe array's page table. When a change occurs, the PAM determines an index of the page and easily converts the index into the value stored in enclave memory. The attacker can easily retrieve leaked data from the kernel module through network, file, or proc file systems, as demonstrated in our study.

The important conclusions from this work are the following: First, this study demonstrated that enclave memory can be leaked rather easily when the SGX application has a memory corruption vulnerability. This contradicts the general belief that SGX's abilities to disallow illegal memory accesses and to prevent injected code from running cannot mitigate the SGX application's memory corruption vulnerability problem. Second, we devised an ROP attack that can loop inside the payload by using gadgets in the enclave. In most cases, once the ROP payload is injected, the gadget's control flow is executed sequentially. However, we successfully implemented the conditional statement by using gadgets of the enclave so that our attack payload can communicate with the outside or execute other attack payload.

Finally, we presented a rather complicated return-oriented programming attack to retrieve the enclave's entire memory. Since it is considered to be hard to construct an ROP payload for complex tasks, people often use ROP attacks for simple but critical operations such as escalating privilege, disarming security mechanism, or calling a desired function. However, we used ROP as the main attack method; the synchronization, data extraction, and loops are implemented through ROP. This does not leave any system logs about security-related configuration changes and function calls, thus making the SGXDump attack more stealthy. We hope the detailed ROP code in this paper will be of great benefit to researchers who are interested in ROP attacks on SGX.

We are trying to build other attack payloads that can utilize a different hardware resource as a communication medium. Since the kernel module can monitor any hardware, we believe that there will be better options than using the page table's access bit.

**Author Contributions:** Formal analysis, H.Y.; Investigation, H.Y.; Software, H.Y.; Validation, H.Y. and M.L.; Writing—original draft, HanJae Yoon; Writing—review & editing, M.L. All authors have read and agreed to the published version of the manuscript.

**Funding:** This research was funded by National Research Foundation of Korea, grant number NRF-2021R1A4A2001810.

**Institutional Review Board Statement:** Not applicable.

**Informed Consent Statement:** Not applicable.

**Data Availability Statement:** Not applicable.

**Acknowledgments:** This work was supported by the National Research Foundation of Korea (NRF) grant funded by the Korea government (MSIT) (NRF-2021R1A4A2001810).

**Conflicts of Interest:** The authors declare no conflict of interest.

## Appendix A. Gadget List in Open Source Project.

**Table A1.** Gadget list in Asylo.

| Gadgets | Offset |
| --- | --- |
| add bl, byte ptr [rcx]; lfence; call rax | 0x1b440b |
| add rcx, rdi; mov rdi, rcx; call rax | 0x28ab7f |
| inc dword ptr [rdx]; ret | 0xa6970e |
| mov qword ptr [rax], rdx; ret | 0x828401 |
| mov rax, qword ptr [rax]; ret | 0x828392 |
| mov rax, rdi; ret | 0x7e4d50 |
| mov rdi, qword ptr [rax]; lfence; call rbx | 0x67fb63 |
| mov rdi, rax; call rbx | 0x050b50 |
| nop; ret | 0x009b68 |
| pop rax; ret | 0x009ac0 |
| pop rbx; ret | 0x404ddc |
| pop rcx; ret | 0x075bdd |
| pop rdi; ret | 0x20d413 |
| pop rdx; ret | 0x07735c |
| pop rsp; ret | 0x40da1d |
| sal ebx, cl; ret | 0x2e8e39 |
| sub esp, dword ptr [rdx − 0x76b80000]; ret | 0x7b6eef |

**Table A2.** Gadget list in Mystikos.

| Gadgets | Offset |
| --- | --- |
| add al, byte ptr [rax]; add byte ptr [rbx + 0x5d], bl; ret | 0x036de3 |
| add rax, qword ptr [rcx − 0x77]; retf | 0x139285 |
| inc dword ptr [rcx − 0x77]; ret | 0x027143 |
| jmp rax | 0x027f2b |
| mov dword ptr [rdi + rdx − 0x27], eax; mov rax, rdi; ret | 0x0b4310 |
| mov eax, edi; ret | 0x0b4315 |
| mov qword ptr [rax], rdx; ret | 0x828401 |
| mov rdi, qword ptr [rax]; lfence; call rbx | 0x67fb63 |
| nop; ret | 0x009b68 |
| pop rax; ret | 0x108665 |
| pop rax; ret | 0x009ac0 |
| pop rbx; ret | 0x03475d |
| pop rcx; or eax, dword ptr [rax]; leave; ret | 0x0136eb |
| pop rdi; ret | 0x034401 |
| pop rdx; add byte ptr [rdx], cl; add cl, cl; ret | 0x028fd2 |
| pop rsp; ret | 0x40da1d |
| shl edi, cl; add byte ptr [rax], al; lfence; call rax | 0x3ea393 |
| sub rax, rcx; ret | 0x1bda72 |

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
