# Peer review of "SGXDump: A Repeatable Code-Reuse Attack for Extracting SGX Enclave Memory"

_applsci, doi:10.3390/app12157655_

Round 1

Reviewer 1 Report

The research proposed an ROP payload that can loop inside the payload by using gadgets in the enclave. To show the effectiveness of our method, the research implemented SGX-Dump which can extract an enclave’s entire memory. 

Suggestions:

1. section 7. Related work should be the "7. Discussion". the related work should be in section 2.

2. In section 2. Background and section 3. Threat Model. In all the section, the research only cited one journal "SGX-Shield [20] " in line 100. It is not reasonable. Not all the section is proposed by the research.

3. In "Figure 1. The SGXDump framework". If the framework is prosed by the research, it should be "The proposed SGXDump framework" will be better.

4. Line 394, "Table 1. Gadget list used in mbedTLS-SGX." It is not explained or cited in the research.

Author Response

Response Letter

(Re.: Manuscript ID: applsci-1818235)

 “SGXDump: A Repeatable Code-Reuse Attack for Extracting SGX Enclave Memory”

by

HanJae Yoon , SungJin Park , ManHee Lee

We, authors, would like to thank the associate editor, guest editors, and the reviewers for their time and valuable comments, and these are helped us to improve the contribution of our manuscript. We have made a great effort to reflect all comments of the reviewers in the revised manuscript. The major changes described in the manuscript are listed as follows:

  1. section 7. Related work should be the "7. Discussion". the related work should be in section 2.
  2. In section 2. Background and section 3. Threat Model. In all the section, the research only cited one journal "SGX-Shield [20] " in line 100. It is not reasonable. Not all the section is proposed by the research.
  3. In "Figure 1. The SGXDump framework". If the framework is prosed by the research, it should be "The proposed SGXDump framework" will be better.
  4. Line 394, "Table 1. Gadget list used in mbedTLS-SGX." It is not explained or cited in the research.

Please find a detailed response to the reviewer’s comments.

Review 1.

  1. section 7. Related work should be the "7. Discussion". the related work should be in section 2..

Response: Thank you for pointing this out. We changed the section title to Discussion.

  1. Background and section 3. Threat Model. In all the section, the research only cited one journal "SGX-Shield [20] " in line 100. It is not reasonable. Not all the section is proposed by the research.

Response: Thank you for pointing this out. We added three more references to related paragraphs.

  1. In "Figure 1. The SGXDump framework". If the framework is prosed by the research, it should be "The proposed SGXDump framework" will be better.

Response: Thank you for pointing this out. We changed the title as suggested.

  1. Line 394, "Table 1. Gadget list used in mbedTLS-SGX." It is not explained or cited in the research.

Response: Thank you for pointing this out. We added the following sentence in section 6.1.

Table 1 shows the gadgets we used in our attack.

Reviewer 2 Report

Intel Software Guard Extensions (SGX) is a set of instructions that allows a user process to create a secure area called an enclave inside its address space. In this paper, it is proposed an ROP payload that can loop inside the payload by using gadgets in the enclave. Hereafter, my comments:

Lines 24-30] You described micro-architectural timing attacks but not controlled-channel attacks. Could you include also the latter?

Line 31] Could you explain in which category ROP attacks can be included? Micro-architectural timing attacks or controlled-channel attacks?

Line 34] Please clarify BLX.

Line 126] The concept of "kernel module installation" must be a expanded to better explain what this module should do. Does it correspond to PAM?

Lines 128-129] Please explain better "the enclave has a memory corruption vulnerability (e.g., a stack buffer overflow)". Are you referring to an application expected to use the enclave?

Line 130] What do you mean via this assumption "We also assume that target enclave is not encrypted at build time"?

Section 3] In general, section 3 should be expanded. Even a diagram/drawing could help the reader to understand the threat model.

Section 5.1] Please clarify how much the memory assumptions are realistic.

Section 5.2] How did you practically find the ROP gadgets?

Section 5.4] It is not clear where the memory data is then exfiltrated.

Lines 421-426] Are these attacks related to the SGX technology?

Conclusion] Which other attack payloads could be executed potentially? And how?

Author Response

Response Letter

(Re.: Manuscript ID: applsci-1818235)

 “SGXDump: A Repeatable Code-Reuse Attack for Extracting SGX Enclave Memory”

by

HanJae Yoon , SungJin Park , ManHee Lee

We, authors, would like to thank the associate editor, guest editors, and the reviewers for their time and valuable comments, and these are helped us to improve the contribution of our manuscript. We have made a great effort to reflect all comments of the reviewers in the revised manuscript. The major changes described in the manuscript are listed as follows:

  1. Lines 24-30] You described micro-architectural timing attacks but not controlled-channel attacks. Could you include also the latter?

Response: Thank you for the query. We added an explanation about the controlled-channel attack in the 2nd paragraph in Section 1 as follows:

A controlled-channel attack is analyzing memory access patterns or memory contents through channels that an attacker can control such as page faults. [9–14].

  1. Line 31] Could you explain in which category ROP attacks can be included? Micro-architectural timing attacks or controlled-channel attacks?

Response: Thank you for the query. We realized that the previous explanation about ROP caused confusion. In the 2nd paragraph in Section 1, we inserted the following sentence for better understanding of ROP.

ROP can be used at the stage of injecting the attack code in both micro-architectural timing attacks and controlled-channel attacks, which are two representative methods of attacking SGX.

  1. Line 34] Please clarify BLX.

Response: Thank you for the query. We changed the sentence as follows:

Furthermore, [16,17] showed that BLX, a branch instruction of ARM CPU, ~

  1. Line 126] The concept of "kernel module installation" must be a expanded to better explain what this module should do. Does it correspond to PAM?

Response: Thank you for the query. Yes, it corresponds to PAM, but in the threat model section, we simply expanded the goal of the kernel module as follows:

Thus, we assume that the attacker can install a kernel module for checking the access bit of the victim process

  1. Lines 128-129] Please explain better "the enclave has a memory corruption vulnerability (e.g., a stack buffer overflow)". Are you referring to an application expected to use the enclave?

Response: Thank you for the query. We changed the sentence for clearer explanation:

we assume that the target application running the enclave has a memory corruption vulnerability (e.g., a stack buffer overflow)

  1. Line 130] What do you mean via this assumption "We also assume that target enclave is not encrypted at build time"?

Response: Thank you for the query. We added more explanation for better understanding.

In some cases, the binary is encrypted at build time to further improve the security of the target enclave. This binary is decrypted just before the enclave is loaded, so it is practically impossible to make ROP payloads because pre-anlysis of enclave binary is not feasible. Therefore, we assume that target enclave is not encrypted at build time.

  1. Section 3] In general, section 3 should be expanded. Even a diagram/drawing could help the reader to understand the threat model.

Response: Thank you for the suggestion. We added a figure describing the threat model.

  1. Section 5.1] Please clarify how much the memory assumptions are realistic.

Response: Thank you for the query. For better understanding, we added the following paragraph in the end of Section 5.1.

The first step is to get the memory mapping information of the target process by analyzing the /proc/pid/maps file as shown in the Figure 3. After checking the mapped memory area in lines 1~6 and the enclave memory area in lines 15~20, the attacker can find memory areas located outside of the enclave. The second step is to identify every thread that accesses the possible probe array memory area by using the debugger. Then, the attacker can select a memory area that there is no threads accessing the area. In this research, we often selected one from the stack area since it normally satisfied the above conditions.

  1. Section 5.2] How did you practically find the ROP gadgets?

Response: Thank you for the query. For better understanding, we added the following sentence in the 2nd paragraph of Section 5.2.

During the attack preparation step, the adversary will search for a set of gadgets. We use ROPgadget[26] for finding all gadgets that can be used to build ROP payloads.

  1. Section 5.4] It is not clear where the memory data is then exfiltrated.

Response: Thank you for the query. For better understanding, we added the following sentence in the 3rd paragraph of Section 5.4.

Right-shifting 0x7f000 twelve times generates the final piece of protected data, 0x7f, originally stored in enclave memory address at 0x1ba6d000. The PAM keeps collecting data via the probe array until the attacker can retrieve entire memory data from the target enclave.

  1. Lines 421-426] Are these attacks related to the SGX technology?

Response: Thank you for the query. Yes. They are related to the SGX technology. So we clarified that by adding SGX in the two sentences.

In MemJam [32], they tried to recover the 128-bit AES key from SGX.

In CopyCat [9], they tried to recover the 2048-bit RSA key from SGX.

12/ Conclusion] Which other attack payloads could be executed potentially? And how?

Response: Thank you for the query. Yes. We inserted one paragraph at the end of Section 8.

We are trying to build other attack payloads that can utilize a different hardware resource as communication medium. Since the kernel module can monitor any hardware, we believe there will be better options than using the page table’s access bit.

Round 2

Reviewer 1 Report

Suggestions:

1. The author on the system and version 1 is :"HanJae Yoon , SungJin Park , ManHee Lee *", but in version 2 is: "HanJae Yoon 1, ManHee Lee 2*". Please explain why to delete the second author.

2. In section 2. Background's subsection:Virtual Address Translation. All the subsection do not cite any journal. This is a background section, it is not reasonable. Please make sure it.

3. if you refer to other journals over all of the research, please make sure you have cited them. There are some sections not cited with any journal. 

Author Response

Response Letter

(Re.: Manuscript ID: applsci-1818235)

 “SGXDump: A Repeatable Code-Reuse Attack for Extracting SGX Enclave Memory”

by

HanJae Yoon , ManHee Lee

We, authors, would like to thank the associate editor, guest editors, and the reviewers for their time and valuable comments, and these are helped us to improve the contribution of our manuscript. We have made a great effort to reflect all comments of the reviewers in the revised manuscript. The major changes described in the manuscript are listed as follows:

  1. The author on the system and version 1 is :"HanJae Yoon , SungJin Park , ManHee Lee *", but in version 2 is: "HanJae Yoon 1, ManHee Lee 2*". Please explain why to delete the second author.
  2. In section 2. Background's subsection:Virtual Address Translation. All the subsection do not cite any journal. This is a background section, it is not reasonable. Please make sure it.
  3. if you refer to other journals over all of the research, please make sure you have cited them. There are some sections not cited with any journal.

Please find a detailed response to the reviewer’s comments.

Review 1.

  1. The author on the system and version 1 is :"HanJae Yoon , SungJin Park , ManHee Lee *", but in version 2 is: "HanJae Yoon 1, ManHee Lee 2*". Please explain why to delete the second author.

Response: As you may know in Korea there is some controversy over the soundness of MDPI, the publisher of this journal. We, Hanjae and Manhee, think this journal is still a good place to publish, but Sungjin thinks differently. So he told us that it is ok to withdraw his name from authors' list. So we had to change the author list.

  1. In section 2. Background's subsection:Virtual Address Translation. All the subsection do not cite any journal. This is a background section, it is not reasonable. Please make sure it.

Response: Thank you for pointing this out. We added more references like below.

Virtual Address Translation. As a computer operates, virtual addresses need to be translated into physical addresses. For this purpose, the OS uses a per-process page table to manage its memory at the page level. The page table is an array of page table entries (PTE). A PTE contains the physical address of a virtual page and additional information including status bits such as accessed, present, rw, user, and dirty among others. From these status bits, we make use of the access bit. When an instruction accesses a page, its access bit is set to one. Later, the access bit is used by the page replacement algorithm to select which page will be swapped out to the secondary disk [28,29].

Newly added references.

  1. Talluri, M.; Hill, M.D.; Khalidi, Y.A. A new page table for 64-bit address spaces. In Proceedings of the Proceedings of the Fifteenth ACM Symposium on Operating Systems Principles, 1995, pp. 184–200.
  2. Arpaci-Dusseau, R.H.; Arpaci-Dusseau, A.C. Operating systems: Three easy pieces; Arpaci-Dusseau Books LLC Boston, 2018.

  1. if you refer to other journals over all of the research, please make sure you have cited them. There are some sections not cited with any journal.

Response: Thank you for pointing this out. We added more references like below.

We demonstrate successful ROP attacks that could lead to leak of the entire memory by using SGXDump and PAM to attack two well-known SGX open source projects: mbedTLS-SGX and Graphene-SGX. While mbedTLS-SGX is a very useful tool for SGX applications that need to use an SSL library for secure communication [24]. Graphene-SGX is a tool for running unmodified code on SGX [25]. Therefore, any vulnerable code found in either of these projects could be serious threats for SGX service owners and users.

Added references

  1. Aublin, P.L.; Kelbert, F.; O’keeffe, D.; Muthukumaran, D.; Priebe, C.; Lind, J.; Krahn, R.; Fetzer, C.; Eyers, D.; Pietzuch, P. TaLoS: Secure and transparent TLS termination inside SGX enclaves. Imperial College London, Tech. Rep 2017, 5.
  2. Tsai, C.C.; Porter, D.E.; Vij, M. Graphene-sgx: A practical library OS for unmodified applications on SGX. In Proceedings of the 2017 USENIX Annual Technical Conference (USENIX ATC 17), 2017, pp. 645–658.

Intel SGX. Intel Software Guard Extensions (SGX) is a set of instructions that allows a user process to create a secure area called an enclave inside its address space. An enclave has separate code and data sections including a stack and heap, SGX safeguards this enclave against external access [5,6]. However, as many previous studies presented, enclave’s memory is still vulnerable to various attacks and the code-reuse attack like ROP is shown to be effective to leak the enclave’s memory [14,20]. In our study, we build the ROP payload that has a loop format by using gadgets inside the enclave. It should be noted that, SGX SDK libraries are statically linked to an enclave’s code at build-time so that the enclave becomes a self-contained binary. This property may hamper an attacker’s ROP composition because the target enclave binary should be available to the attacker beforehand. However, this is not necessarily true because SGX is designed to secure enclaves under a compromised OS and the OS has the privilege to look into the enclave binaries [5,6].

Added references.

  1. Costan, V.; Devadas, S. Intel SGX Explained. IACR Cryptol. ePrint Arch. 2016, 2016, 1–118.
  2. Intel. Intel(R) Software Guard Extensions (Intel(R) SGX) SDK for Linux OS, "2020". "Accessed: 2022-07-12".
  3. Lee, J.; Jang, J.; Jang, Y.; Kwak, N.; Choi, Y.; Choi, C.; Kim, T.; Peinado, M.; Kang, B.B. Hacking in darkness: Return-oriented programming against secure enclaves. In Proceedings of the 26th USENIX Security Symposium (USENIX Security 17), 2017, pp. 523–539.
  4. Schwarz, M.; Weiser, S.; Gruss, D. Practical enclave malware with Intel SGX. In Proceedings of the International Conference on Detection of Intrusions and Malware, and Vulnerability Assessment. Springer, 2019, pp. 177–196.

Figure 1 shows the overall of threat model. The primary goal of SGXDump is to extract the memory’s entire contents including any code and data in the target enclave by using ROP payload. To this end, we first suppose that the attacker is able to acquire root privilege, so has complete control of the OS. This is not an outlandish assumption because Intel SGX’s programming model is designed to secure the enclave under the assumption that any of the software or hardware components, except the enclave, could be compromised [30]. Thus, we assume that the attacker can install a kernel module for checking the access bit of the victim process.

Newly added reference

  1. Zheng, W.; Wu, Y.; Wu, X.; Feng, C.; Sui, Y.; Luo, X.; Zhou, Y. A survey of Intel SGX and its applications. Frontiers of Computer Science 2021, 15, 1–15.

As explained in the Intel® Software Guard Extensions (Intel® SGX) SDK for Windows OS, SGX does not allow dynamic linking, meaning that all necessary functions’ binaries are copied to the enclave code at build time [6]. Each enclave needs its own function sets at different locations. This is why it is almost impossible to construct a silver bullet ROP payload that will work for any enclave applications. Therefore, an adversary has to build a customized ROP payload for each target enclave.

Added reference

  1. Intel. Intel(R) Software Guard Extensions (Intel(R) SGX) SDK for Linux OS, "2020". "Accessed: 2022-07-12".

Identifying the Enclave’s Base Address. To find out the base address of the Enclave, we make use of the /proc/pid/maps file. This file contains the address information of the /dev/isgx file, which has the device name of the SGX kernel module [34]. The address is identical to the base address of the loaded Enclave in the target process.

Newly added reference

  1. Bowden, T.; Bauer, B.; Nerin, J.; Feng, S.; Seibold, S. The/proc filesystem. Linux Kernel Documentation 2000.

Reviewer 2 Report

Thanks for the improvement. I think the paper deserves to be published.

Round 3

Reviewer 1 Report

Ok. no more problem.